Towards an easier creation of three-dimensional data for embedding into scholarly 3D PDF (Portable Document Format) files

Newe Axel axel.newe@fau.de
Chair of Medical Informatics, Friedrich-Alexander University Erlangen-Nuremberg , Erlangen , Germany
Thompson Steven
Electronic publication date: 2015 Mar 3
Publication date: 2015
Volume: 3
Electronic Location ID: e794
Received 2014 Dec 3; Accepted 2015 Feb 4
Copyright: © 2015 Newe
Copyright year: 2015
Copyright holder: Newe
License: This is an open access article distributed under the terms of the Creative Commons Attribution License, which permits unrestricted use, distribution, reproduction and adaptation in any medium and for any purpose provided that it is properly attributed. For attribution, the original author(s), title, publication source (PeerJ) and either DOI or URL of the article must be cited.
License URL: https://creativecommons.org/licenses/by/4.0/

Keywords: Portable Document Format, Universal 3D, PDF, 3D, Model, U3D, Geometry, Point cloud, Line set, Mesh

Funding: The author declares there was no funding for this work.

==============================
The Portable Document Format (PDF) allows for embedding three-dimensional (3D) models and is therefore particularly suitable to communicate respective data, especially as regards scholarly articles. The generation of the necessary model data, however, is still challenging, especially for inexperienced users. This prevents an unrestrained proliferation of 3D PDF usage in scholarly communication. This article introduces a new solution for the creation of three of types of 3D geometry (point clouds, polylines and triangle meshes), that is based on MeVisLab, a framework for biomedical image processing. This solution enables even novice users to generate the model data files without requiring programming skills and without the need for an intensive training by simply using it as a conversion tool. Advanced users can benefit from the full capability of MeVisLab to generate and export the model data as part of an overall processing chain. Although MeVisLab is primarily designed for handling biomedical image data, the new module is not restricted to this domain. It can be used for all scientific disciplines.

Introduction

Science produces a large amount of three-dimensional (3D) data, especially in the biomedical field. Hence, the visualization of this data should be 3D as well in order to avoid a loss of information (Tory & Möller, 2004). However, almost all contemporary visualization means (paper printouts, computer or television screens) only provide a two-dimensional (2D) interface. This limitation makes it necessary to project the 3D data onto the available 2D plane, which results in the so-called “2.5D visualization” (Tory & Möller, 2004). While there are many technical approaches to overcome this issue (stereoscopy, anaglyphs, head-mounted displays), they all require special equipment and do not eliminate the problem that objects can occlude each other if they lie on the same optical axis. A simple but effective solution is interaction: by changing the angle of a 2.5D projection, depth perception is improved (Tory & Möller, 2004), and masked objects can be brought to sight. Finally, the possibility to interact with the data representation might trigger the consumer’s curiosity, which then may lead to a more detailed exploration (Ruthensteiner & Heß, 2008).

The Portable Document Format (PDF) is the de-facto standard for the exchange of electronic documents with more than 500 million users worldwide (Adobe Systems Incorporated, 2014a). A PDF file describes the layout of an electronic document and can comprise all necessary resources for its reproduction without the need of external resources—including 3D models.

Although this technology has been available since 2007, and although even journals encourage authors to embed 3D data directly into their publications ((Maunsell, 2010) and http://www.elsevier.com/about/content-innovation/u3d-models), the usage of 3D PDF has not yet found its way into the publishing routine.

One reason might be that the generation of the necessary data can still be challenging, especially for the inexperienced. This constitutes an inhibition threshold, which stands in the way of a further proliferation of 3D PDF usage in scholarly communication. In this article, a new tool for the generation of 3D data for embedding into PDF is presented. It is available free of charge for all major operating systems, can be used out-of-the-box for biomedical and other science disciplines, and requires no programming skills.

Background and Related Work

Using 3D PDF for the communication of biomedical and other scientific data

The Portable Document Format is a standard (ISO 32000-1:2008, (International Organization for Standardization (ISO), 2008)) for the definition and reproduction of electronic documents. It ensures independence from the creating, displaying and printing hardware and software and fulfills all requirements for an interactive document as defined by Thoma et al. (2010). The specification is freely available and very well documented. Although it is an ISO standard, it can be downloaded and used free of charge (Adobe Systems Incorporated, 2014b; Adobe Systems Incorporated, 2008).

The Adobe Reader (http://get.adobe.com/reader/otherversions/) is currently the only 100% standard compliant software that allows to correctly display 3D models (meshes with solid or transparent surfaces, wireframes, line sets and point clouds) embedded into PDF documents and to let the user interact with them (zooming, panning, rotating, selection of components). By means of embedded scripting, animations and interaction with other components (e.g., control elements like clickable buttons) of the respective PDF document are possible. Adobe Reader is available for free for all major operating systems (MS Windows, Mac OS, Linux).

Embedded 3D models were proven suitable and useful for electronic publication by several authors in a variety of scientific disciplines. In 2008, the first application in biomedical science was published, showing anatomic details of two species obtained from histological images as 3D figures (Ruthensteiner & Heß, 2008). The first medical applications followed in 2011, presenting dental molds (Danz & Katsaros, 2011) and the skin of a human face (Ziegler et al., 2011). Simulated volume rendering in PDF documents has been demonstrated by Ruthensteiner, Baeumler & Barnes (2010) for three different imaging methods. 3D PDF was used to describe anatomical (Quayle et al., 2014) and functional (Van de Kamp et al., 2014) characteristics of animal species. Finally, also chemical (Vasilyev, 2010), molecular (Kumar et al., 2010) and astronomical (Barnes & Fluke, 2008) use cases are described.

3D model data

Two different file formats can be used to embed 3D models into PDF: the Product Representation Compact (PRC) format and the Universal 3D (U3D) format. They are not arbitrarily exchangeable since they differ in some features (Visual Technology Services Ltd., 2014a; Visual Technology Services Ltd., 2014b). PRC (Adobe Systems Incorporated, 2014c) is the older format and published as ISO 14739-1 (International Organization for Standardization (ISO), 2012), but U3D seems to have become more popular. The academic publishing company Elsevier, for example, invites authors to enrich their articles by supplementary 3D models in U3D format (http://www.elsevier.com/about/content-innovation/u3d-models) and many 3D software tools provide the possibility to export in U3D format rather than PRC. The U3D specification is published by the Ecma International as the ECMA-363 (Universal 3D File Format) standard (ECMA International, 2007). The most important details about the U3D file format are described in Newe & Ganslandt (2013).

There are a number of tools and libraries available for the creation of either PRC or U3D files. Most of them are commercial software, but open source solutions like MeshLab (http://meshlab.sourceforge.net/) are available. However, using these tools needs a considerable amount of training. Furthermore, certain features those are of interest for scientific 3D data like point clouds and line sets (polylines) are often neglected by these tools. Free libraries that provide the full feature range are available, e.g., the Visualization and Computer Graphics Library (http://vcg.sourceforge.net/) or the U3D Reference Implementation Library (http://u3d.sourceforge.net/). Barnes et al. (2013) have presented a comprehensive library that creates PRC data for scientific purposes. The drawback of using such libraries, however, is that they often require programming skills. Furthermore they are only suitable for converting existing 3D data that need to be generated intermediately using other software like Amira (http://www.vsg3d.com/amira/).

MeVisLab

An alternative to 3D data processing applications and libraries is the use of MeVisLab (http://www.mevislab.de/, German web address but English language) since it combines the advantages of both. MeVisLab is an image processing framework, developed by MeVis Medical Solutions AG and Fraunhofer MEVIS in Bremen, Germany. The core feature of MeVisLab is its module concept: All included algorithms and functions are represented and accessed by “modules”, which can be arranged and connected to image or data processing networks on an intuitive graphical user interface (GUI) following the visual data-flow development paradigm. These networks can then be converted with little effort into complete applications for a convenient re-use. For simple tasks like creating or converting 3D model data, this requires no writing of software code.

MeVisLab is available for all major operating systems (MS Windows, Mac OS and Linux, http://www.mevislab.de/download/) and offers a free license for use in non-commercial organizations and research (“MeVisLab SDK Unregistered”, (MeVis Medical Solutions & Fraunhofer, 2014a).

The current version 2.6.1 of MeVisLab has included more than 1,000 pre-defined standard modules. Another 1,800 additional modules completely wrap the Insight Segmentation and Registration Toolkit (ITK, http://itk.org/) and the Visualization Toolkit (VTK, http://vtk.org/), which makes the total module base very comprehensive.

Motivation

The standard distribution of MeVisLab comes with the WEMSaveAsU3D module, which is described in detail in Newe & Ganslandt (2013). This module is capable of exporting triangular surface meshes to U3D file format and has proven to be suitable for use in reporting of surgery planning results in clinical routine (Newe, Becker & Schenk, 2014). However, it was limited to triangular meshes—other geometry definitions could not be exported, although U3D allows for embedding point clouds and line sets (polylines) as well.

Most publications regarding the application of 3D PDF for scholar communication focus on three-dimensional surface models defined by those triangular meshes. While this might be the most impressive use case, point clouds and polylines also play an important role that has so far been underrepresented in the literature. One of the very few examples is the illustration of the distribution of galaxies by means of a point cloud (Barnes & Fluke, 2008). Polylines have only been used as a means for labeling (Barnes & Fluke, 2008) although they could add value in many other use cases, e.g., for the depiction of vectorcardiograms (VCGs). Figure 1 shows two 2.5D projections of VCGs, originally published in Sur, Han & Tereshchenko (2013). In order to compensate the limitation to a single projection, the authors of that publication created movies (WMV format: http://dx.doi.org/10.1371/journal.pone.0057175.s001and http://dx.doi.org/10.1371/journal.pone.0057175.s002) in which the figures are rotated in different directions, since they had no other option to demonstrate the 3D complexity.

Figure 1 2.5D projections of vectorcardiograms.

These VCG illustrations were published in Sur, Han & Tereshchenko (2013) and are restricted to one single projection. The original figure image is licensed under CC-BY 4.0 (http://creativecommons.org/licenses/by/4.0/).

Finally, the definition of object model trees and the re-use of geometry data for multiple objects was not possible using WEMSaveAsU3D and the specification of object properties (like name, color etc.) was somewhat cumbersome. In summary, no free software tool or library was available so far that allowed for creating all three types of 3D model data (point clouds, polylines, meshes) for embedding into PDF without the need of programming.

Methods

Creation of new modules

In order to compensate these shortcomings, two new modules for MeVisLab were created. Core of the development was the new “SaveU3D” module. The source code was written in C++ using Microsoft Visual Studio 2010 and contains the full set of constant definitions (e.g., lighting attributes or material attributes) of the ECMA-363 Standard. Although not necessary for the end-user, the code was extensively annotated in order to simplify the understanding and the potential expansion of the implementation.

To facilitate the generation of the necessary object definitions, the GUI of the final module was enriched by a convenience window that allows for a comfortable assembly of the respective information. This part was written in the Python programming language.

In order to support the conversion of existing point cloud and line set data, an additional C++ module named “MarkerListImport” was developed.

Validation of the U3D export module

The new SaveU3D module was validated against the old WEMSaveAsU3D module and the reference implementation of the U3D standard (http://u3d.sourceforge.net/).

Using only features that the WEMSaveAsU3D module already supported (triangle mesh geometry, lighting, cameras), the output files of the SaveU3D and WEMSaveAsU3D were equal.

The output files of the reference implementation and the SaveU3D module were binary compatible, i.e., both produced the same geometry models after import into a PDF document.

Results

The SaveU3D module

The new module produces ECMA-363 compliant U3D files that contain point clouds, line sets, triangle meshes and meta data annotations. Each object that shall be exported can be specified using a simple description syntax that resembles the well-known Extensible Markup Language (XML) (Fig. 2A). A Specification Generator (Fig. 2B) facilitates the creation of the necessary specifications, so that learning the description syntax for specifying e.g., names and colors of objects is not even necessary. Objects can be grouped and placed into a hierarchical object tree. Finally, the same geometry data can now be used for multiple objects of the U3D scene.

Figure 2 Panels (Windows) of the SaveU3D module.

(A) Main panel of the U3D module where the object specifications are entered. (B) Panel of the Specification Generator which facilitates the assembly of valid object specifications.

While the objects to be exported are specified directly in the GUI of the module, the underlying geometry data must be provided by other modules. This can be modules that load existing data from a storage medium (see description of the second new module, MarkerListImport, below as an example) or modules that create the data themselves. This concept gives the user the full flexibility to either re-use data that has been created using other software or to use MeVisLab itself to generate the data.

Usage of the module

Using the module is straightforward: it simply needs to be created (“instantiated”) using the MeVisLab GUI. A detailed description of the module and its usage is available with the module documentation. An example network that demonstrates all features is also available. The module documentation and the example network can be accessed by right-clicking any instance of a SaveU3D module and selecting “Show Help” or “Show Example Network” from the context menu.

Two steps are required to export 3D objects: first, the geometry data must be connected to the inputs of the module, and second, the objects must be specified using the module GUI. The specifications can be entered manually using the XML-like Syntax or by using the Specification Generator. The latter can be opened by right-clicking the module and selecting “Show Window” → “Specification Generator Panel” from the context menu. Deleting unwanted or erroneous specifications can only be done directly in the GUI. The specification syntax is described in the module documentation.

Point clouds

Point clouds are the simplest type of objects. They only consist of points in 3D space. Therefore, the geometry data only consists of a list of point positions that need to be connected to the “inPointPositions” input of the module.

Line sets

Line sets consist of nodes that are connected by edges. The geometry data must be provided by two lists: the positions of the nodes (“inLinePositions” input) and the description of the edges (“inLineConnections” input), that indicates which nodes (positions) are connected by an edge. The latter list can be omitted—in this case the connections are automatically calculated by simply connecting each node with the next in the list.

Meshes

Meshes are the most complex type of objects since they consist of nodes, edged, faces, and some more attributes. MeVisLab uses the Winged Edge Mesh (WEM) format (Baumgart, 1972; Baumgart, 1975) to store meshes. Therefore, the geometry data for meshes needs to be provided as a WEM and connected to the “inWEM” input. MeVisLab provides about 50 modules for the import, creation, manipulation and visualization of WEMs.

The MarkerListImport module

The second new module (Fig. 3) enables the user to load plain text files like Character-Separated Values (CSV) files that contain tabular point cloud data or line set data and to attach these data directly to the SaveU3D module. The example network of SaveU3D and Fig. 3B demonstrate this. MarkerListImport has some basic flexibility since it allows for the specification of number delimiters and decimal separators. It also allows for filtering the input file, thus enabling the user to import only a subset.

Figure 3 Panel (Window) of the MarkerListImport module and example of use.

(A) The main panel of the import module needs only a few essential parameters. (B) This simple network is all that is needed to import a point cloud or a line set and to export it in U3D format.

Availability

The source code (http://sourceforge.net/p/mevislabmodules/code/HEAD/tree/trunk/Community/General/Sources/ML/MLPDF/ and http://sourceforge.net/p/mevislabmodules/code/HEAD/tree/trunk/Community/General/Modules/ML/MLPDF/) of the SaveU3D module is available at the MeVisLab Community Sources (http://mevislabmodules.sourceforge.net/), hosted by Sourceforge.net. It is part of the MLPDF project and requires another community project (MLBaseListExtensions). The MarkerListImport sources can be found in the MLBaseListExtensions sub-folders. Binary versions of the modules for the current MeVisLab version 2.6(.x) and instructions how to add them to an existing MeVisLab installation are provided as Supplemental Information 1. Binaries for future releases of MeVisLab can be requested from the author or the MeVisLab community.

Discussion

A tool that facilitates the creation of 3D data

While the predecessor module WEMSaveAsU3D (Newe & Ganslandt, 2013) was already a big relief compared to other solutions, the new SaveU3D module simplifies the generation of biomedical 3D models for embedding into PDF even further. In addition, new geometry model types (point clouds and line sets) can now be exported using MeVisLab in the same simple and user friendly way as meshes.

Line sets can be used for the visualization of nervous fiber tracking, vessel centerlines or as pointers to structures of interest. The example use case and motivation for this article, however, is the interactive visualization of vectorcardiograms. Traditional methods of plotting VCGs onto a 2D surface as in Fig. 1 come with an inherent loss of information unless a high number of plots with different viewing angles or some other kind of workaround like a movie (Sur, Han & Tereshchenko, 2013) is presented. Using Adobe 3D PDF technology, only one representation of the VCG data needs to be included in a document, and the user herself or himself can find the best way to display the respective graph according to his or her question. Figure 4 and Supplemental Information 2 illustrate the same data as Fig. 1 in this fully interactive way (Adobe Reader required).

Figure 4 Example of two 3D vectorcardiograms exported by the SaveU3D module.

Shows the same 5-beat vectorcardiograms as Fig. 1 as displayed in Adobe Reader. A coordinate system and additional fiducial markers at the x-axis amplitude are integrated into the 3D scene. See also Supplemental Information 2. (Click images to enable interactive mode in Adobe Reader.)

Finally, point clouds (or single points) can be used to mark or highlight points of interest (Fig. 4 and Supplemental Information 2). To stay with the example of VCGs: points or point clouds could be used to mark the amplitude values or to set fiducial markers.

This support for point clouds and line sets is a unique feature of the SaveU3D module for MeVisLab, compared with other integrated tools like MeshLab. Some libraries do support point clouds and line sets (e.g., S2PLOT (Barnes et al., 2013)), but they require programming. The SaveU3D module works “out-of-the-box” and without the need to write any software code. The generation of the geometry data can be handled within MeVisLab or be loaded from external sources, alternatively. Therefore, this solution provides the possibility to make use of both approaches: on the one hand, the SaveU3D module could be used as a simple conversion tool (Figs. 3, 5 and Supplemental Information 3); on the other hand, it can be used as final step in an arbitrarily complex image or data processing pipeline.

Figure 5 Example of 3D surface meshes converted by MeVisLab using the SaveU3D module.

(A) The exported meshes as displayed in Adobe Reader. One mesh has been exported using color and transparency. (Click image to enable interactive mode in Adobe Reader.) (B) The MeVisLab network used to load and to export the five meshes. The original mesh data was downloaded in Wavefront OBJ file format from BodyParts3D, The Database Center for Life Science (Mitsuhashi et al., 2009) and is licensed under CC-BY-SA 2.1 Japan. See also Supplemental Information 3.

In addition to the new geometry types and the Specification Generator tool, some minor improvements were integrated as well. The SaveU3D module allows for defining hierarchical object trees. These object trees can be used to apply functions of embedded scripting code to single objects or to groups of objects (e.g., single axes or the whole coordinate system of Fig. 4). Since geometry data can be used for multiple objects now, the size of a U3D file can be significantly lower now if an object shall be available in different shapes (e.g., colors, rendering modes).The auxiliary MarkerListImport module provides the possibility to directly and conveniently load and convert existing point cloud or line set data (Fig. 3B). Experienced users can take this module as a basis for creating own import modules with possible extended or modified functionality.

A novice MeVisLab user will certainly need some basic training, but that applies for any new software. The usage of MeVisLab is explained in its comprehensive documentation (http://www.mevislab.de/developer/documentation/). The “Getting Started Tutorial” is especially recommended for newcomers. It can be accessed via MeVisLab itself (Menu “Help” → “Show Help Overview” → “Getting Started”) or as downloadable version (MeVis Medical Solutions & Fraunhofer, 2014b). Further support can be obtained from an active online community (http://forum.mevis.fraunhofer.de/index.php).

U3D models produced with the SaveU3D module can be processed by free PDF authoring tools like LaTeX (http://www.latex-project.org/) in conjunction with the media9 package (http://www.ctan.org/pkg/media9) or like the iText library (http://itextpdf.com/) to create the final PDF document.

File size

A general issue regarding the embedding of 3D data into PDF files is the resulting file size (Ruthensteiner & Heß, 2008; Kumar et al., 2010; Newe & Ganslandt, 2013). For point clouds and line sets, the difference between a static 2.5D projection image and a U3D file is nearly negligible or even beneficial in favor of the native 3D data. The two high resolution bitmap sub images of Fig. 4 in print quality (820 × 827 pixels, 300 dpi, 7 × 7 cm2) have a file size of 90–95 kB in lossless compressed PNG format. The respective U3D files have a size of only 55–72 kB. In contrast, the movies used as workaround in Sur, Han & Tereshchenko (2013) have a total size of more than 30 MB. The U3D files would grow with additional data, but the general dimension will roughly stay the same.

The size of meshes ranges in different scale. The U3D data for Fig. 5A has a size of 453 kB, which is about 3 times the size of a bitmap image in print quality. However—if it is assumed that at least three 2.5D projections of the depicted objects must be presented to get a rough impression of their spatial extent, the smaller size of the projection images becomes almost insignificant.

As regards the total size of PDF documents with embedded 3D models, a comparison between a number of published scholarly articles and a preprint of this article reveals no significant size issue. The twelve articles published by PeerJ (https://peerj.com/articles/) in the second calendar week of 2015 have an arithmetic mean size of 2,009,111 bytes (std. dev. 1,775,301 bytes). The (below print-quality) preprint of this article (https://peerj.com/preprints/560v1.pdf) with embedded models for a VGG and vertebral bodies has a total size of 1,223,399 bytes, which is only about 60% of the mean size of the published articles. Nevertheless, mesh data can be much larger and therefore the creator of the mesh model has to make a tradeoff between model details and file size. Considering modern wide area network speed, file sizes of around 10 MB should not be a problem. At this scale, even comprehensive results of liver surgery planning with multiple resection proposals can be exchanged using 3D PDF technology (Newe, Becker & Schenk, 2014).

Limitations

With respect to precision and fidelity of biomedical and other scientific data, it is strongly recommended to use Adobe Reader to display 3D PDF documents containing this kind of data, since it is the only software that is absolutely compliant to the PDF standard. Other viewers might work, but that is not guaranteed. However, while the Adobe Readers for Windows, MacOS and Linux fully support the 3D models embedded into PDF, the currently available versions for iOS and Android do not. Therefore, as regards tablet computers, 3D PDF documents can be displayed as intended on x86/64 based devices with Windows operating system, but not on the popular Apple iPad or on devices that use ARM processors. All PDF readers that are available for the latter two platforms are not capable of displaying embedded 3D models.

A major feature that is still missing is the application of textures onto models. Using textures would allow for creating simulated volume rendering visualizations. Although this simulated volume rendering would be limited to a fixed windowing, it would offer a sophisticated way of publishing biomedical 3D images.

Another interesting feature is animation as presented in Van de Kamp et al. (2014), but since the definition of animations (which may require to determine axes, rotations, translations, speeds) can be a quite complex task that needs many sub steps, an implementation seems not reasonable for the SaveU3D module which is intended to be kept as simple as possible.

Unfortunately, PDF does not support another feature of U3D: glyphs. Glyphs could be used to apply label text directly within the 3D scene, but since PDF ignores U3D glyphs, it is not planned to add support for glyph to SaveU3D. However, using textures for labeling objects in the 3D scene can serve as a workaround as shown in Barnes et al. (2013). PDFs with an educational purpose (e.g., for teaching anatomy to medical students) could especially benefit from an undoubtful labeling of structures with complex spatial relationships.

Ongoing development

SaveU3D is part of the MLPDF project and will be extended. In the long term, this project shall provide all modules that are needed to create PDF documents using MeVisLab. One of the next steps is the addition of the missing features mentioned above and the implementation of a module that exports 3D objects in PRC format. Finally, the creation of actual PDF files is on the agenda for future development.

Another MeVisLab project deals with the implementation of the DICOM Surface Segmentation Storage SOP Class (DICOM Standards Committee, Working Group 6, 2008). As soon as the respective modules for loading and saving DICOM Surface Segmentation objects are available, the gap between generic DICOM segmentation results and their conversion into files for embedding into PDF will be closed.

Beyond biomedical data and scholarly communication

Although it is their primary purpose, MeVisLab or the SaveU3D module are not limited to biomedical data. The 3D objects used for Fig. 5A could also be representations of e.g. chemical molecules (Kumar et al., 2010; Vasilyev, 2010). Line sets or meshes could illustrate geological surfaces in order to replace or complement isolines. Point clouds as used in Fig. 4 could be derived from physics models or statistical distributions of any science discipline. Graf (2012) used 3D PDF to illustrate a dipole field and two examples from astronomy can be found in Barnes & Fluke (2008).

Finally, scholarly communication is only one application, PDFs with embedded 3D models can be used for. Several authors have shown that interactive 3D visualization improves anatomy learning (Jurgaitis et al., 2008; Petersson et al., 2009). 3D PDF is an advanced development of the technology used by them and provides much more control for the users. Therefore, it could be a very good tool for training of students or future specialists.

Conclusion

Modern science in general and the biomedical sciences in particular produce three-dimensional data at an increasing rate. The Portable Document Format provides the means to distribute and visualize this data without the need for additional dedicated software. The new modules for MeVisLab presented in this article enable even novice users to generate the necessary file format without requiring programming skills and without the need for an intensive training. Advanced users can benefit from the full potential of MeVisLab to generate and export the model data to U3D as part of the overall processing chain. Only one additional PDF authoring tool is then required to generate the final PDF document. For most uses cases, a replacement of this authoring tool does not seem reasonable since the ability to set text layout, process screenshots etc. is an application domain of its own. Therefore, in most cases, the separation of model data generation and PDF authoring cannot be avoided anyway.

Supplemental Information

Supplemental Information 1 Binary files, module definition files and installation instructions. Using these files, the SaveU3D module and the MarkerListImport module can be added to an existing MeVisLab 2.6(.x) installation without the need to compile the source files.

Click here for additional data file.

Supplemental Information 2 Two example U3D files of the vectorcardiograms shown in Figure 4 and the resulting 3D PDF.

Click here for additional data file.

Supplemental Information 3 An example U3D file of lumbar vertebrae as shown in Fig. 5 and the resulting 3D PDF.

Click here for additional data file.

Supplemental Information 4 U3D data for Fig. 4

Contains two zipped U3D file for embedding the interactive 3D scenes into Fig. 4.

Click here for additional data file.

Supplemental Information 5 U3D data for Fig. 5

Contains a zipped U3D file for embedding the interactive 3D scene into Fig. 5.

Click here for additional data file.

De-identified VCG data by courtesy of Larisa G. Tereshchenko, Knight Cardiovascular Institute, Oregon Health & Science University, Portland, Oregon, USA.

Abbreviations

2D Two-dimensional

2.5D Three-dimensional reduced to two-dimensional

3D Three-dimensional

CSV Character-separated Values (also: Comma-separated Values)

DICOM Digital Imaging and Communications in Medicine

GUI Graphical User Interface

ITK Insight Segmentation and Registration Toolkit

PRC Product Representation Compact file format

SOP DICOM Service-Object-Pair

U3D Universal 3D file format

VCG Vectorcardiogram

VTK Visualization Toolkit (VTK)

XML Extensible Markup Language

Additional Information and Declarations

Competing Interests

Author Contributions

Data Deposition

The author declares there are no competing interests.

Axel Newe conceived and designed the experiments, wrote the paper, prepared figures and/or tables, reviewed drafts of the paper, engineered the software.

The following information was supplied regarding the deposition of related data:

Sourceforge: http://sourceforge.net/p/mevislabmodules/code/HEAD/tree/trunk/Community/General/Sources/ML/MLPDF/

http://sourceforge.net/p/mevislabmodules/code/HEAD/tree/trunk/Community/General/Modules/ML/MLPDF/.

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
