# Peer review of "Towards an easier creation of three-dimensional data for embedding into scholarly 3D PDF (Portable Document Format) files"

_PeerJ, doi:10.7717/peerj.794_

## Round 0.1 · original submission · Minor Revisions

Please note Reviewer 1's concerns over grammar and semantic word choices such as "exchange" vs. 'shared," and minor sentence issues that keep this paper from being publishable as is. While they are an easy fix, note comments and recommendations for other sections by Reviewer 1 that should be appropriately addressed when valid for the author.

A greater concern is over the suitability of this paper in its current form for the journal as I agree with Reviewer 1 that it appears to be more of an instruction manual or technical tutorial rather than a unique contribution to the scientific community. There is not enough theoretical methodology as would be expected and the rhetorical situation seems posited in information rather than persuasion. While the primary claim may be supported with practical evidence throughout, more theoretical research claims with valid support are expected. This paper could stand on its own as a how-to document with removal of the word "scientific" or its replacement with an alternate adjective, since use of the word "scientific" therein is relatively superfluous at times; it reads a little bit like 'tacked on' rather than a core, imperative element.

Reviewer 2 considers the paper well-written; yet, if it is intended to reveal results of an experiment with improvement to the MeVisLab software, that procedure would qualify it more for something like an upgrade sheet or errata rather than a brand new journal contribution.
While I believe the subject matter is relevant to scientists, and more so to any users of the software desiring proposed interactivity in the PDF format, I would like to see more emphasis on potential dynamics of that interactivity in a revision. These issues should be adequately addressed moving forward if the paper is to be made suitable for publication in PeerJ.

·

Basic reporting

This paper describes an improved module for the MeVisLab system that, in addition to mesh-based models supported by previous versions, allows point clouds as well as poly-lines to be exported in the U3D format. These models can subsequently be embedded in PDFs which, using Adobe Reader, can be viewed as interactive 3D objects.

The article describes a piece of software rather than an experiment, so shoehorning it into a structure that contains sections like 'materials and methods' feels a little contrived. Overall the meaning is clear enough, though in places the prose is a bit cumbersome and could do with some smoothing. The style isn't always appropriate for a scientific article, and sometimes feels as though it's turned into an advert or tutorial rather than an objective description of a piece of software.

Overall as a piece of software, I can see that this is a valuable contribution. In the interest of turning this review around quickly, I've not had a chance to try the software out for myself (in spite of the claims of 'easy to use', learning to use any modelling software from scratch is a non trivial job!) but there are enough examples included in the supplementary materials for me to take the fact that it works at face value. I'm less convinced by the 'big picture' that's woven around the description of the software, and can't help feeling that this would be stronger simply as a 'here is a useful thing' paper.

It would be good to see some form of evaluation, even if the primary purpose of the article is to announce the existence of a piece of software. For example, there's no consideration of the size of the models created or the impact that this would have on typical PDF file sizes.

It's probably worth citing DOI:10.1371/journal.pone.0102355 as related work.

I think the title of this paper is rather too broad and perhaps too grand: it implies a fundamentally new technique for embedding 3D models into PDFs, whereas actually what's described is a relatively modest improvement to a existing module in an existing software package to generate models in the style already supported by the PDF format. In terms of scope of 'scientific three-dimensional models', the abstract makes it clearer that the technique is primarily applied in the context of 'biomedical image data', and specifically in terms of the MeVisLab framework, but the title suggests something more profound (and while I accept that MeVisLab can be extended to deal with other forms of scientific data than those it already supports, I'd argue that writing a new module in C++ to cater for one's favourite kind of scientific data doesn't quite fit into the 'easy generation' claim of the title). Personally I'd go for something a bit more focussed and prosaic, such as "A method for embedding 3D biomedical models in PDFs using MeVisLab" (though on reflection even this wouldn't be quite right, since what the paper really talks about is a method for creating U3D models that using other tools can be embedded in PDFs). As well as better describing the content of the article, something along these lines would be easier to find in search engines and also removes the need for the 'towards' hedging.

I have to take issue with the first sentence of the abstract which states that "The Portable Document Format ... is therefore particularly suitable to exchange and present respective data, especially as regards scholarly articles". Given that it's still by far the most popular way of distributing scientific articles, I'd say that in the context of scientific articles, the PDF is an acceptable way for exchanging prose that's to be read by humans; and it's not a bad way for exchanging some kinds of diagrams that are to be looked at by humans (though even in both these cases it has its limitations). At a push I might go with the notion that it's a suitable way to present some aspect of some data to support some story that you're trying to tell in the main body of the paper. But in the general case, it's a pretty awful way to exchange data, and I'd strongly resist anything that encourages embedding any kind of data in a PDF for the purpose of data-exchange. I'm okay with the idea that a PDF can be usefully used to *present* a kind of visual thumbnail of some underlying data, and that this can be a valuable addition to static images, but that's not at all the same as exchanging or sharing it which is better done by making it available via some appropriate repository (e.g. Figshare or Dryad). In any case it's a little misleading to say that this is scientific data that's being shared; since the PDF format only supports a couple of 3D modelling formats, what's actually being embedded is a transformed representation of the underlying data, stored in terms of points and polygons. This is fine as a way for a human reader to get a better sense of what the author is talking about by allowing some basic interaction with a 3D model but isn't the same as giving the reader the ability to access and analyse the data for themselves. In the version of Acrobat I have, there doesn't even appear to be a mechanism for extracting the object from the PDF, so to all intents and purposes, even the 3D representation is locked inside the format and isn't available for the reader to look at/analyse in other packages. It's probably worth noting that the same principle/gripe applies to 2D figures and tables in PDFs; they are essentially limited interpretations of underlying data for the purposes of telling a story, and not substitutes for sharing the actual data.

11 The sentence 'Although the vizualisation of this data should be 3D as well to avoid a loss...' is confusing. After a few readings I realised that the 'as well' bit refers to the previous sentence and the 'although' to the second clause, but that took a bit of work on my part. I'd suggest rewording the first two sentences to something like

"Science produces a large amount of three-dimensional (3D) data, especially in the biomedical field, however almost all contemporary visualization means (paper printouts, computer or television screens) only provide a two-dimensional (2D) interface which results in a loss of information."

24 Same issue as previously with the claim that the PDF "is the best way to exchange respective data"; this issue runs throughout the paper in different guises.

26 "Although there is a considerable number" should be "there are a considerable number" (the determiner "a number of" is used with plural nouns).

30 "has not yet found its way neither into the publishing routine nor into the clinical routine." could more tightly be "not yet found its way into either the publishing or clinical routines"

56 The PDF/H approach was new to me, and I'm not quite sure what to make of it in the context of scholarly publishing. The idea seems to be that in healthcare, the PDF can be used as a secure container for electronically shuttling both structured (XML) and unstructured (images and scanned documents) between healthcare practitioners. I can see that in this context, using the PDF as a kind of secure electronic envelope has some advantages over posting paper copies around a complex healthcare system, but I'm not convinced it's a sensible approach to data sharing in the general case. In particular in the context of scientific articles, as the datasets become larger, embedding them in a PDF would tend to create massively bloated PDFs, making them less attractive to download and store. The article describes PDF/H as a "standard", though this is explicitly contradicted by Kohn (http://www.aiim.org/documents/standards/PDF-H/PDF_Healthcare_Article.pdf) which says "PDF Healthcare is NOT a proposed standard". It's a bit hard to figure out exactly what PDF/H is; the site and related documents contain several dead links (http://www.aiim.org/standards/article.aspx?ID=33284) or reports that have to be paid for , and there doesn't seem to have been much activity since 2009, so I'm not sure how relevant or widely used this approach is.

96 This section reads somewhat like an advert for MeVisLab; I'm uncomfortable about claims such as "easy-to-understand" documentation.

156 This section feels like a tutorial rather than something I'd expect to see in a scientific article.

218 I'm not sure if it's possible to select the thumbnail shown for an 3D model here or whether this is created automatically from the underlying data. Although the 3D version works fine in Adobe Reader, in other PDF readers (e.g. OSX's Preview) there's a lot of wasted grayness in this figure, which also appears to be overly compressed with a lot of compression artefacts. Either the figure captions or the main body text should make it clear that instructions such as 'Click the image to enable interactive mode' only apply if you're reading the article using a recent version of Adobe Reader.

239 The comment about being available for different operating systems has already been made on line 96.

244 This section feels a bit like a braindump of plans and ideas; it would be good somewhere to have an understanding of how this project is funded and the timescales involved, if for no other reason than to give potential users some sense of whether it is likely to be maintained for long enough to warrant the effort of learning to use it.

265 The section on 'Beyond Biomedical Data' lacks substance and is unconvincing. I don't know anything about geological data, but it's really not clear that this approach could be used sensibly for chemical molecules beyond the very trivial; because the technique relies on polygonal meshes etc, representing anything beyond small molecules would create massive datasets that would bloat the PDF beyond reason (typically the data for larger molecules such as proteins are shared in formats that model the atoms and their interconnectivity, with the 3D meshes or other representations being generated on the fly by specialist molecular-rendering software; even these compact representations can run to Mb of data, and a mesh representation in U3D would be orders of magnitude bigger).

279 "seems not reasonable" -> "does not seem reasonable" ?

Experimental design

There is no experiment here, so experimental design isn't relevant.

Validity of the findings

Again, there aren't really any 'findings' as such -- the paper describes a piece of software.

·

Basic reporting

The author describes the development and function of a module that creates U3D files to be embedded in PDF documents. The module (SaveU3D) is an improvement against an old module (WEMSaveAsU3D) also developed by the author and already published. The author claim is that the new module supposedly has some advantages not only when compared with the old module but also when compared to other modules and libraries because it would be easier to use and does not require any programing skills.

Experimental design

No comments

Validity of the findings

The procedure seems to be well validated.

Additional comments

The results show codes written with ingenuity and good organization level. As the paper is a very technical one, it is difficult to review it in terms of a classical scientific hypothesis. Then the best thing I could do was to use the module and compared with the other modules and libraries. So I went to the “field” and tested MeshLab, VCG and the U3D reference implementation library. It was a very time consuming task but also pleasant. I agree with the author that SaveU3D is relatively simple to use but I am not really sure if it requires less training than using MeshLab for example. A certain amount of good training is also necessary with the author’s module. As an example is the use of the XML-like script language to export the objects. The problem is that this is an opinion of ONE novel user. The final opinion will be the one from all the users that experience SaveU3D over the other modules. However it is also true when the author pointed out that the “support for point clouds and line sets is a unique feature of the SaveU3D module” compared to MeshLab.

---

## Round 0.2 · accepted · Accept

Reviewer 1 has made valid comments on review of the last author revision and I concur with the first point made about the transfer of 'exchange' to forms of 'communicate' having previously noted how aspects of the term 'share' -- while not the best choice -- may have been worth further consideration. I say this because I was somewhat startled to see the final word choice in the revision, also, so we both struggled in review regarding the best exact term here. With that said, no further revisions are required for acceptance though the author is prompted to consider the overall value of comments in most positively moving forward.

·

Basic reporting

See general comments

Experimental design

See general comments

Validity of the findings

See general comments

Additional comments

I'm happy that my initial comments have largely been addressed, and suggest the paper is now in a publishable state. There are a few things that I think could be improved still, but which I have to admit I'm struggling to make practical or constructive suggestions about.

1) The use of the word 'communicate' instead of exchange/share definitely reduces my unease about the idea that this is a way good way of moving *data* from one place to another, but it doesn't completely remove the issues mentioned previously. However, I can't see how to fix that without a substantial re-writing, and I don't think that the effort is warranted and would be a lot of work to fix what is after all a somewhat tangential point.

1. The title, and the implication throughout that this is both an easy and generic approach still doesn't quite sit true. The U3D format itself is undoubtedly generic, but that's prior work and not the subject of the paper. The MeVisLab software can only really be defined as 'easy to use' when it's used for the kind of data it already deals with well, and I'm fairly sure that using it in the context of other data would require considerable effort. So there is some collision here between the idea of it being a 'general purpose solution' and being 'easy'. But this is somewhat of a philosophical point that's hard to untangle and I don't think needs get in the way of publication.